# Dosimetric Comparison Study of Proton Therapy Using Line Scanning versus Passive Scattering and Volumetric Modulated Arc Therapy for Localized Prostate Cancer

**DOI:** 10.3390/cancers16020403

**Published:** 2024-01-17

**Authors:** Masaru Takagi, Yasuhiro Hasegawa, Kunihiko Tateoka, Yu Takada, Masato Hareyama

**Affiliations:** 1Department of Radiation Oncology, Sapporo Teishinkai Hospital, Sapporo 065-0033, Japan; 2Department of Radiation Physics, Sapporo Teishinkai Hospital, Sapporo 065-0033, Japan

**Keywords:** prostate cancer, pencil beam scanning, line scanning, passive scattering, treatment planning, side effect

## Abstract

**Simple Summary:**

The currently documented outcomes of proton therapy for patients with localized prostate cancer are based on passive scattering (PS). In this study, we compared the dose distribution of the line scanning (LS) method, one of the pencil beam scanning (PBS) techniques, with that of PS and volumetric modulated arc therapy (VMAT) in 30 patients. The results showed that LS could significantly reduce the wide range of rectal and bladder doses compared to the PS and VMAT while upholding the requisite target dose. The results suggest that proton therapy for prostate cancer with the PBS method may have the potential to reduce toxicities further while preserving therapeutic efficacies.

**Abstract:**

Background: The proton irradiation modality has transitioned from passive scattering (PS) to pencil beam scanning. Nevertheless, the documented outcomes predominantly rely on PS. Methods: Thirty patients diagnosed with prostate cancer were selected to assess treatment planning across line scanning (LS), PS, and volumetric modulated arc therapy (VMAT). Dose constraints encompassed clinical target volume (CTV) D98 ≥ 73.0 Gy (RBE), rectal wall V65 < 17% and V40 < 35%, and bladder wall V65 < 25% and V40 < 50%. The CTV, rectal wall, and bladder wall dose volumes were calculated and evaluated using the Freidman test. Results: The LS technique adhered to all dose limitations. For the rectal and bladder walls, 10 (33.3%) and 21 (70.0%) patients in the PS method and 5 (16.7%) and 1 (3.3%) patients in VMAT, respectively, failed to meet the stipulated requirements. The wide ranges of the rectal and bladder wall volumes (V10-70) were lower with LS than with PS and VMAT. LS outperformed VMAT across all dose–volume rectal and bladder wall indices. Conclusion: The LS method demonstrated a reduction in rectal and bladder doses relative to PS and VMAT, thereby suggesting the potential for mitigating toxicities.

## 1. Introduction

Radiotherapy is one of the radical treatments for patients with localized prostate cancer. The curability of radiotherapy for localized prostate cancer is comparable to that of surgery [1,2]. Various methods of radiation therapy coexist, including X-ray therapy, brachytherapy, proton therapy, and carbon beam therapy. In common with these radiotherapy methods, to achieve both high curability and low rates of toxicity, it is essential to concentrate the dose on the target volume and reduce the dose to surrounding organs at risk (OARs). High-dose X-ray therapy using intensity-modulated radiation therapy (IMRT) is currently considered one of the standard radiotherapies for prostate cancer [3].

Given the physical characteristics of the Bragg peak, proton therapy can, in principle, better reduce the dose to OARs than X-ray-based techniques [4]. Proton therapy has been used to treat localized prostate cancer for the past 20 years. Recently, several studies have reported proton therapy results for localized prostate cancer, all of which showed favorable biochemical control and low rates of late gastrointestinal (GI) and genitourinary (GU) toxicities [5,6,7,8,9]. It is significant to note, however, that these results were all obtained with the technique of passive scattering (PS).

The current proton therapy technology is gradually shifting from PS to pensile beam scanning (PBS) methods that were developed to achieve more efficient dose delivery of protons [10]. PBS is a new proton irradiation method that has rapidly been adopted worldwide. Proton therapy’s fundamental concept was proposed in the 1940s [11]. In the 1990s, the first clinical proton therapy system was introduced in a hospital [12]. The PS technique was used in early facilities to provide proton treatment. The technical foundation for the PBS method, a novel proton irradiation technique, was developed in the 1980s [13]. The PBS method was used for the first time in Japan in the 2010s and is currently available throughout many regions, including Europe and the US [14]. As of 2023, 11 out of 19 facilities in Japan have started to perform proton therapy using the scanning method.

In theory, the PBS method, including line scanning (LS), can provide more localized dose distribution than the PS method, which is expected to improve prostate cancer treatment outcomes. However, due to the indolent nature of prostate cancer, the long-term outcomes of the PBS method have not yet been established. In addition, comparative studies of treatment plans involving either the scanning method or the PS method are limited.

In this study, we conducted a comparative study of treatment plans based on the hypothesis that the LS method for localized prostate cancer would reduce the dose to the rectum and bladder compared with the PS method and volumetric modulated arc therapy (VMAT).

## 2. Materials and Methods

### 2.1. Patients

Thirty patients who received proton therapy between January 2017 and March 2019 were selected at random for participation in this study. This study followed the standards of the Declaration of Helsinki and the current ethical guidelines and was approved by the XXXX Institutional Review Board (R030221). All patients were fully informed about this study and provided signed written consent forms.

### 2.2. Computed Tomography (CT) Simulation and Contouring

CT imaging was performed using a SOMATOM Perspective CT Scanner (SIEMENS, Munich, Germany), with each patient in the supine position with a full bladder. The patients’ trunks were immobilized using NB Bord ENB-16 immobilization shells (Engineering System Co., Ltd., Nagano, Japan). Their lower bodies were fixed by using an air mattress (Vac-Lock, CIVCO Medical Solutions, Kalana, IA, USA) to avoid fluctuations around the femoral head during treatment. CT images of 1.25 mm thickness were obtained. Magnetic resonance images were obtained in 3 mm slices in all patients. The bladder and rectum were aimed at in order to maintain the same conditions as in the planning CT. T2wi and T1Gd images were fused to the CT for treatment planning.

After the CT and MRI images were superimposed, contouring of the clinical target volume (CTV) and OARs was performed. The CTV was defined as the entire prostate and part of all seminal vesicles, depending on the risk classification. The 3 mm inner volumes of the entire bladder and the rectum within 10 mm in the SI direction of the CTV were defined as the bladder and rectal wall, respectively, and were used for evaluation in this study.

### 2.3. Dose Constraints

The prescribed dose for the CTV was standardized to 76.0 Gy or Gy RBE (relative biological effectiveness) in 38 fractionations. The RBE for proton therapy was defined as 1.1 for Co-60. The dose constraint was the same for all three treatments and was stipulated as follows: (1) CTV: D50 ≥ 76.0 Gy (RBE), D98 ≥ 73.0 Gy (RBE); (2) rectal wall: V65 < 17%, V40 < 35%; (3) bladder wall: V65 < 25%, V40 < 50%. Dx is defined as the dose received by x % of the organ volume. Vx is defined as the relative volume of the organ that received at least x Gy or Gy RBE.

In cases where the CTV dose constraints could be determined, the dose to the OARs was reduced. Our hospital started proton therapy using the wobbler method in 2016; in 2017, we changed to the scanning method. The definitions of contouring and dose constraints used in this study were those applied in the actual treatment of patients.

### 2.4. Treatment Equipment

The proton therapy system used (Sumitomo Heavy Industries, Ltd., Tokyo, Japan), which has an energy range of 70–230 MeV, was equipped with a universal nozzle that can switch between the PS and LS methods. Truebeam STx (Varian Medical Systems, Palo Alto, CA, USA) was used for the VMAT treatment. All three treatment plans were generated using Version 13.7 of Eclipse (Varian Medical Systems, Palo Alto, CA, USA).

#### 2.4.1. LS Method

The LS method is one of the scanning methods used in proton therapy [15]. The equivalent depth of water from the body surface to the target in the direction of the treatment beam is calculated, and then the deepest slice is swept with the optimal proton energy. Subsequently, the proton energy is changed, and the dose is delivered to a shallower slice of it. These steps are repeated for all of the target cross-sections to form a spread-out Bragg peak (SOBP) [16].

Along with spot scanning and raster scanning, the LS method is one of the most commonly used PBS techniques [17]. These three methods differ in scan time, which may cause clinical problems in tumors with respiratory movement [18]. However, prostate cancer does not require consideration of the respiratory motion. We conducted this analysis with the assumption that the dose distribution of the LS technique might be representative of PBS.

The most crucial characteristic of proton beams is that they stop in the area/region they are supposed to, unlike penetrating X-rays. This characteristic, meanwhile, also adds to treatment uncertainty. Thus, range uncertainty must be considered in the treatment planning of proton therapy, particularly for the PBS method [19]. In this research, the LS method of treatment planning is based on robust optimization, which takes into account the change in the CTV and the range uncertainty of the proton beam. It produces a dose distribution that takes into account the internal and setup margins, virtually the same as the PTV [20,21].

The treatment plan used two beams with gantry angles of 90° and 270°, with the isocenter set in the center of the CTV. A pencil beam algorithm was used with a 2.5 × 2.5 × 2.5 mm dose grid for the proton dose calculation. Finally, for the obtained beam information, dose calculations were performed for a total of 12 different displacements of ±5 mm (3 mm on the rectal side) relative to the isocenter and ±3.5% uncertainty in the CT values and stopping-power ratio conversion to evaluate the CTV dose.

#### 2.4.2. PS Method

The PS proton beam was irradiated from the cyclotron accelerator and then expanded to the required SOBP using beam-wobbling magnets, a lead scatterer, and ridge filters. Compensators were designed and applied to adjust the distal shape of the SOBPs according to the target locations and the direction of the beams.

The PS beams were centered on the PTV with gantry angles = 90° and 270°. Setup errors of 5 mm were assigned to the CTV, similar to the definition of the PTV. A beam-specific PTV (bs-PTV) was created and planned using a CT value, an uncertainty of ±3.5% in the stopping-power ratio conversion, and smearing margins of 6 mm [22,23]. The irradiation field was designed using multileaf collimators (MLCs) with a field margin of 10 to 12 mm, according to the shape of the bs-PTV. Dose calculations were performed using the pencil beam algorithm with a 2.5 × 2.5 × 2.5 mm dose grid. In the latter eight fractions of PS therapy, the cone-down technique was used to reduce the dose to the rectum. The aggregated dose results from the initial and boost irradiation fields are reported using the PS method.

#### 2.4.3. VMAT

VMAT (RapidArc, Eclipse Treatment Planning System version 13, Varian Medical Systems, Palo Alto, CA, USA) is an advanced form of IMRT. VMAT uses one or more arcs that simultaneously vary the gantry rotation speed, dose rate, and leaf position of the MLC to deliver a highly conformal radiation dose to the target [24].

Two coplanar arcs with a 10 MV X-ray were used in this study. The PTV was set as a 5 mm margin from the CTV, with a margin of 3 mm to that of the posterior. This was based on the assumption that image guidance would be carried out using cone beam CT, and a smaller margin was used compared to that used in several clinical trials. Dose optimization was performed using the photon optimizer 13.7 of the Eclipse treatment planning system. The dose calculation algorithm used was AcurosXB, with a calculation grid of 2.5 × 2.5 × 2.5 mm.

### 2.5. Treatment Plan Comparison and Statistical Analysis

Unlike PS and VMAT, LS does not have the concept of a PTV. However, LS does involve both setup errors and beam uncertainties in its planning and evaluation process. Hence, we consider them similar, although the three treatment planning methods do not use the same evaluation set. Accordingly, in this study, the CTV was used for dose analysis of the target volume. D98, D50, D02, Dmax, and the heterogeneity index (HI) were compared as indices of the CTV. HI is calculated as follows: (D02–D98)/D50. For the rectal and bladder walls, the evaluation indices were every 10 Gy for V10 to V60 and every 5 Gy for V65 to V75, Dmean, and Dmax. All parameters above were evaluated using the Friedman test, followed by adjustment using the Bonferroni method, with *p* < 0.05 considered to be statistically significant. Statistical analysis was performed using SPSS version 22.0 (IBM, Armonk, NY, USA).

## 3. Results

### 3.1. Overall Results

A total of 30 patients were included in this study; risk groups as defined by the National Comprehensive Cancer Network were 5, 10, 10, and 5 patients in the low-, intermediate-, high-, and very-high-risk groups, respectively. The median age was 70.5 years. The median prostate and CTV volumes were 30.5 and 32.2 mL. Further details are summarized in Table 1. A comparison of the dose distribution between the three treatment modalities for a representative example case is shown in Figure 1.

Figure 2 shows the DVH curves for each patient and the median values in the CTV (a), rectal wall (b), and bladder wall (c), respectively, which are summarized in Table 2.

In all 30 patients that were investigated, the LS technique satisfied the dose restrictions for the CTV, rectal wall, and bladder wall. The PS technique failed to fulfill the criteria in 10 (33.3%) and 21 patients (70.0%) in the rectal wall (9 patients in V65 and V40 and 1 in V65) and bladder wall (18 patients in V65 and V40 and 3 in V65), while meeting the CTV criteria. Although all the patients satisfied the requirements for the CTV using the VMAT method, 5 patients (16.7%) failed to meet the dose constraints in the rectal wall (2 patients in V65 and V40 and 3 in V65), and 1 patient (3.3%) failed to meet the dose constraints in the bladder wall (V65).

### 3.2. CTV

All three treatments showed similar dose coverage of the CTV, although the overall index values in LS and PS were slightly more favorable than those in VMAT. PS was better than LS only in terms of Dmax.

### 3.3. Rectal Wall

Figure 3a shows the box-and-whisker plots that summarize the results obtained for the rectal wall. LS reduced the rectal wall dose in the range of V10–V70 and the Dmean with statistical significance over the other two methods. For V65, LS showed a dose decrease of 4.4% compared to PS and 2.1% compared to the VMAT cases. For V40, the corresponding values were 8.8% and 5.8%, respectively.

PS was the best among the three methods in the high-dose range (V75 and Dmax). There was no dose–volume index for which VMAT was better than LS. Comparing PS and VMAT, PS reduced the dose in the low (V10–V20) and high-dose regions (V75 and Dmax), while VMAT reduced the dose in the medium-dose region (V30–V65).

### 3.4. Bladder Wall

LS was the best for all indices except Dmax in the bladder wall (Figure 3b). LS showed a median drop of 9.9% and 1.3% compared to PS and VMAT in V65 of the bladder wall. For V40, the corresponding values were 12.6% and 6.1%, respectively.

As observed in the rectal wall, LS outperformed VMAT in all indicators. Only in Dmax did PS show a lower dose than LS and VMAT. VMAT yielded a lower irradiated volume in the medium- to high-dose range (V30 to V70) compared to PS.

## 4. Discussion

This is the first study to simultaneously compare LS, PS, and VMAT treatment plans for localized prostate cancer. The LS technique was successful in all of the patients examined, even though the rectum and bladder constraints in this study were stricter than those typically used in clinical trials [25,26]. According to this study’s findings, switching from PS proton therapy to LS for localized prostate cancer could further reduce side effects while maintaining the treatment’s effectiveness.

### 4.1. Rectal Dose and GI Toxicities

According to the findings of this study, the level of rectal bleeding observed after the use of the LS method is expected to be comparable to that observed after the use of the PS method and better than that following VMAT. Several studies have reported that late rectal bleeding results from an irradiated rectal volume with a high dose range (from > 60 Gy to Dmax) [27,28]. Colaco et al. analyzed 1285 patients treated with proton therapy and reported that V75 was a prognostic factor [29]. Their results showed that rectal wall V75 < 9.2% was associated with significantly less rectal bleeding of grade 2 or higher. Of the 30 patients we analyzed in this study, all patients fulfilled this V75 criterion in LS, with the highest case representing only 7.7%. The median index of LS was inferior to PS in V75 and Dmax in this study, although the difference was only slight (0.9% and 1.3 GyE).

Along with reducing the occurrence of rectal bleeding, LS proton therapy could also reduce the occurrence of other GI toxicities. The findings of several studies suggest a necessity for overall dose reduction, not only in the high-dose range but also in the low-to-medium dose range, to minimize late GI toxicities. A recent study revealed that other GI symptoms, such as fecal incontinence, changes in bowel movement frequency, rectal pain, and rectal ulceration, are associated with low-to-medium dose ranges [30]. Among these symptoms, fecal incontinence has been reported to be highly influenced by V30-V40 [31,32].

The use of a perirectal hydrogel spacer (SpaceOAR; Augmenix, Waltham, MA, USA) in conjunction with scanning proton therapy will lessen overall GI toxicity as well as rectal bleeding. SpaceOAR is a bioabsorbable hydrogel that is inserted between the rectum and prostate before radiation therapy to create a temporary anatomic separation [33]. This device reduced the incidence of rectal bleeding following IMRT for prostate cancer in a phase 3 trial [34]. Additionally, a recent study showed that using hydrogel spacers during proton therapy can reduce the incidence of rectal bleeding [35]. Combining dose reduction in the medium-to-low dosage range by means of the scanning method with dose reduction in the high-dose range through spacer placement could result in a safer dose escalation.

### 4.2. Bladder and Urethral Dose and GU Toxicities

Our results revealed a general dose reduction in the bladder wall using LS compared to the other two methods. However, it is currently not clear to what extent these results will improve clinical GU toxicities. To summarize the results of previous studies, late GU toxicities do not correlate as clearly with DVH parameters as late GI toxicities do [36,37]. Multiple factors might impede the explanation for the correlation between DVH parameters and late GU toxicities. Unlike late GI symptoms, GU symptoms require a longer time to develop over several years [5,6]. Even without radiation therapy, aging increases the number of patients who experience GU symptoms [38]. In addition, a remaining major problem is that late GU toxicities are symptoms caused by both an irradiated bladder and urethra, and it is essentially difficult to identify which organ is the primary contributor [37]. Herein, the impacts of irradiation on the bladder and urethra on GU toxicities are discussed individually.

It is expected that implementing the LS technique will lead to a reduction in GU toxicities through a decreased bladder dose. Multiple studies have demonstrated an association between bladder dose and GU toxicities, along with various clinical variables [37,39,40]. According to the available research, dose reduction is necessary for the entire bladder as well as the bladder triangle [41,42]. The problem, however, is that the bladder fluctuates during treatment and that the irradiated dose may differ from the planned dose [43]. Based on the evidence presented, bladder dose reduction is likely to partially alleviate GU toxicities.

For GI toxicities, dose reduction in the urethra is as important as in the bladder. Clinical trials using stereotactic body radiotherapy have revealed a relationship between urethral dose and early and late GU toxicities typified by urethral stricture [44,45]. However, compared to traditional whole prostate irradiation, a prospective clinical trial designed to reduce the urethral dose and improve GU side effects revealed worse biochemical control [46]. In addition, routine urethral dosage reduction also faces numerous technical difficulties [47]. Reducing the dose in the urethra, located approximately in the middle of the prostate, is more challenging than a partial dose reduction in the bladder. Given these data, further research is needed on the necessity and feasibility of urethral dose reduction.

### 4.3. Comparison of Proton Therapy and IMRT

The results of this study revealed that all dose volume indices in VMAT were inferior to LS and partially better than PS. Currently, IMRT is one of the standard radiotherapies for patients with localized prostate cancer. VMAT is an advanced form of IMRT, and its main feature is the ability to shorten the time of treatment. Furthermore, some studies have demonstrated that VMAT improves the dose distribution compared to that of IMRT [48]. Proton therapy using PS has been reported to offer favorable disease control and low late GI and GU toxicity rates in several studies, although a direct comparison of the PS method with IMRT has not been reported at the time of publication [5,6,7,8,9].

Based on our findings, it is not inconsistent to find that IMRT and proton therapy using the PS technique did not significantly differ in terms of overall GI and GU toxicities in several studies. The planning comparisons between PS and IMRT have revealed that dose reduction for OARs with proton therapy is mainly observed in the low-to-medium dose range [49,50,51]. In terms of toxicities, several trials comparing proton therapy and IMRT have reported conflicting results [52,53,54]. Vapiwala et al. recently published the results of a multicenter, retrospective study of IMRT and proton therapy using mild hypofractionation in 1850 patients at low and intermediate risk [55]. The incidence of severe late GI and GU toxicities was low and did not differ between the two groups. However, details on the specific IMRT (static or rotational) and proton therapy techniques (PS or scanning) used are not available.

Tran and colleagues conducted a comparative study evaluating intensity-modulated proton therapy, VMAT, and 4π radiotherapy in 10 patients diagnosed with prostate cancer [56]. The results of their investigation suggest that the potential of proton therapy to reduce the radiation dosage in the bladder and rectum is limited to the high-dose range. Their findings are in contrast with those of the current study, which could have been influenced by factors such as the scanning beam’s spot size and the gantry angle selected for the treatment.

### 4.4. Comparison of the LS Method and the PS Method

In the current study, a significant dose reduction in the rectal and bladder walls was observed with LS compared to PS, and this method is expected to reduce the risk of clinical toxicities. However, contrary to our results, the PC001-09 study reported no difference in late GU toxicities at 12 months between the scanning and PS methods [57,58]. In addition, regarding GI toxicities, the scanning method was slightly inferior to the PS at 12 months. It is difficult to directly interpret the inferiority of the scanning method based on the results of this study alone since the published results do not provide information on the patients’ backgrounds, including comorbidities and pretreatment GU symptoms. The scanning method is a novel proton irradiation method with few reports of efficacy issues and toxicity in prostate cancer. Future randomized control trials will need to confirm whether scanning with improved dose distribution reduces late GU and GI toxicities compared to IMRT and PS. In particular, long-term follow-up is necessary to evaluate late GU toxicities since the incidence of late GI toxicities reaches a plateau in the first 2–3 years after proton therapy, while the incidence of late GU toxicities tends to increase [5,6].

At many facilities, proton therapy is transitioning from the PS method to the PBS method, represented by LS. Due to the indolent nature of prostate cancer, it will take a long time to evaluate new treatments. In the future, however, it is expected that clinical reports will confirm that PBS is associated with fewer toxicities and that PBS will become the standard proton therapy method for prostate cancer. On the other hand, proton therapy has weaknesses in terms of cost and availability. VMAT, the current standard of treatment, is also expected to remain an important treatment option.

### 4.5. Strengths and Limitations

The strength of our research is that it is the first study to compare the three treatments for prostate cancer in a large number of patients. However, our study has several limitations. The most significant limitation of this study is that it is an in silico study and did not compare the actual late toxicities that occurred. Long-term follow-up is necessary to evaluate the late toxicities caused by radiation therapy for prostate cancer, especially GU toxicities. Therefore, a prospective trial with a planned follow-up of at least five years is desirable to compare treatment modalities. Considering this study’s results, we are currently preparing a prospective trial to compare the different clinical effectiveness of LS, PS, and VMAT in prostate cancer patients. Second, the target doses had to be compared in terms of CTV because of the different conceptions of PTV in the three irradiation methods. However, the OARs dose for the three irradiation modalities was found to be significantly different even when the minimally needed CTV dose was used. Third, our study does not compare the organs that affect sexual function in each treatment. Finally, cost and availability, which should be considered for treatment options, as well as efficacy and toxicities, were not considered in this study.

## 5. Conclusions

The LS method can further reduce the dose to the rectal and bladder walls while maintaining the dose to the CTV compared to the PS method and VMAT, which may result in a reduced incidence of late GI and GU toxicities.

## Figures and Tables

**Figure 1 cancers-16-00403-f001:**
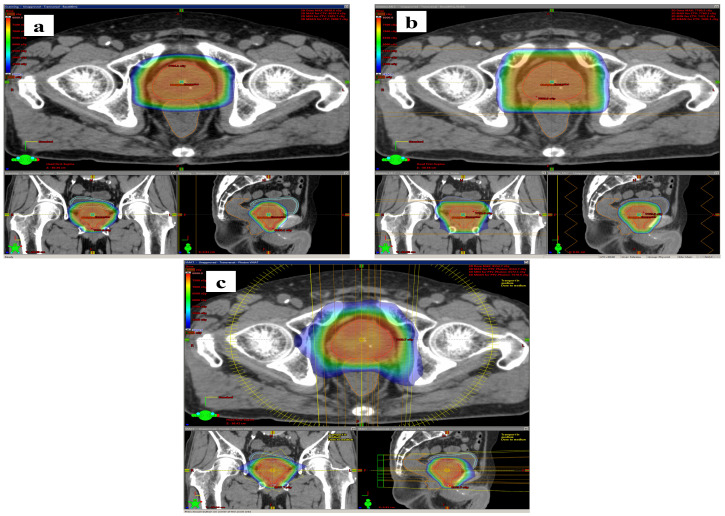
Isodose color wash images of a typical patient planned using line scanning (LS) (**a**), passive scattering (PS) (**b**), and volumetric modulated arc therapy (VMAT) (**c**). Seventy-six-year-old patient with favorable intermediate-risk prostate cancer (cT2a, GS = 3 + 4, initial PSA = 9.82 ng/mL). Prostate volume = 50.5 mL.

**Figure 2 cancers-16-00403-f002:**
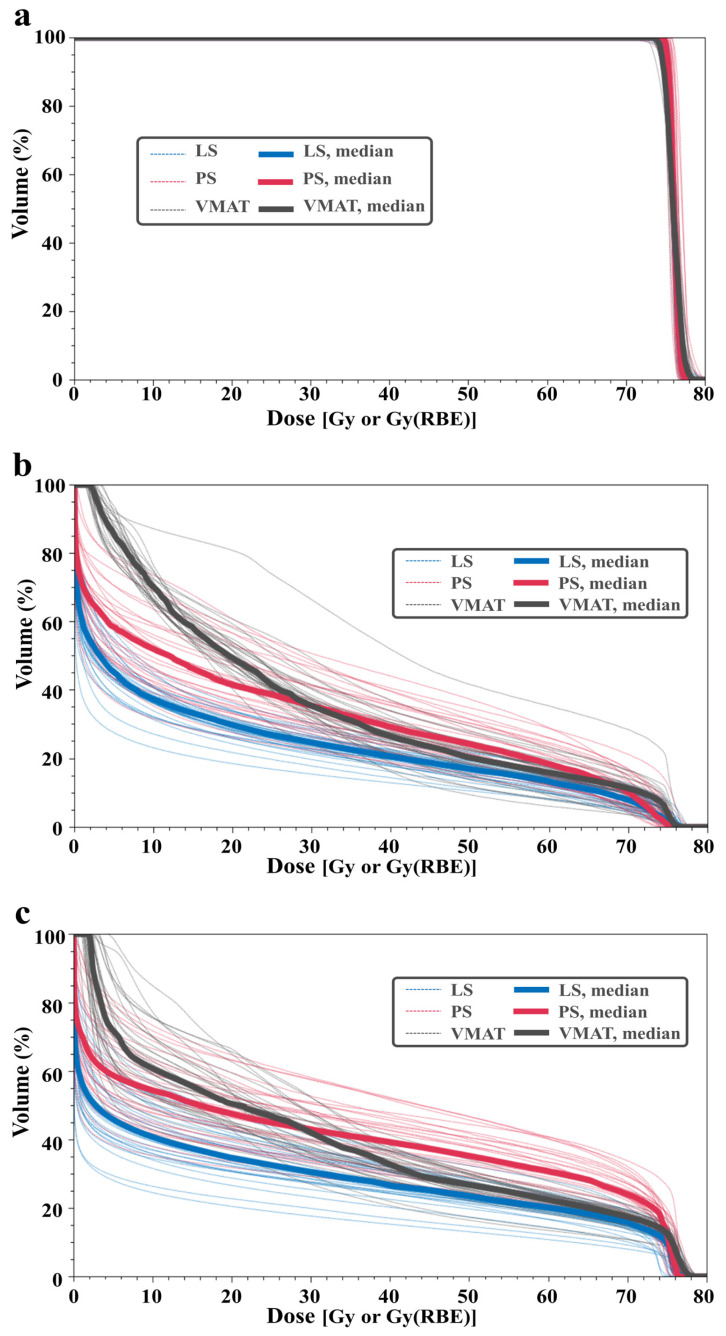
Dose–volume histogram of all patients for the CTV (**a**), rectal wall (**b**), and bladder wall (**c**).

**Figure 3 cancers-16-00403-f003:**
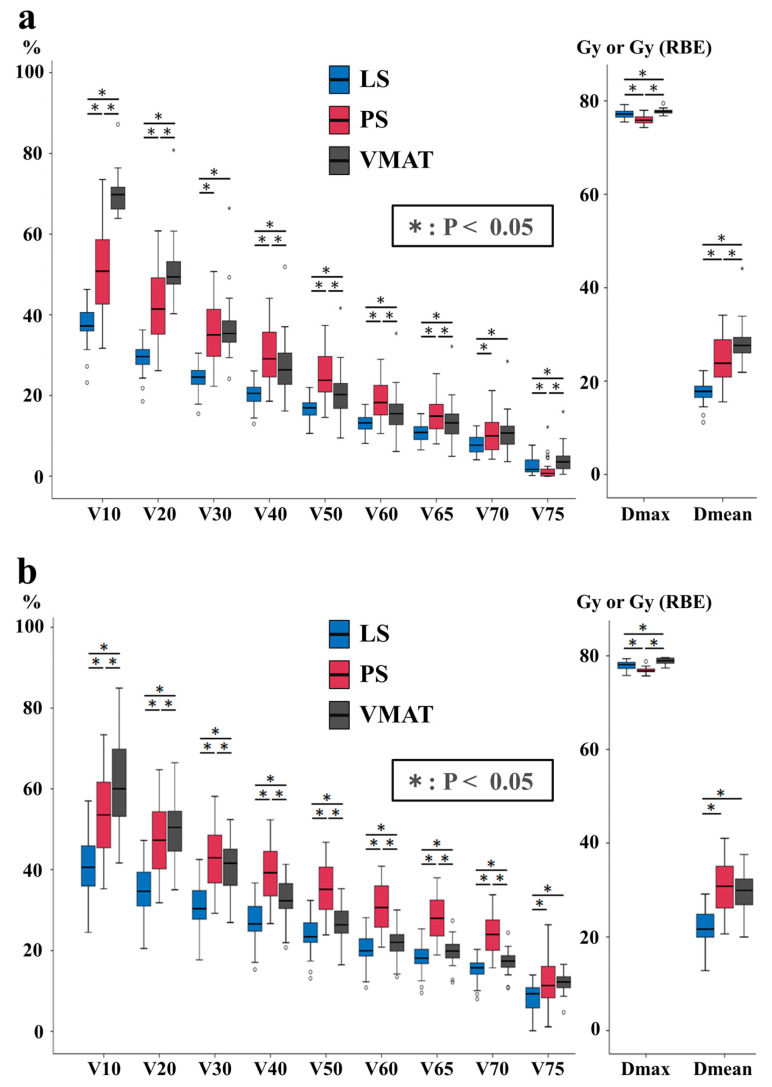
Box-and-whisker plots of dose–volume metrics of the rectal wall (**a**) and the bladder wall (**b**).

**Table 1 cancers-16-00403-t001:** Patient characteristics.

Variable	Level	N	%
Number		30	100
T classification	T1c	14	46.7
	T2a/b/c	5/2/5	16.7/6.7/16.7
	T3a/b	3/1	10/3.3
Gleason score	6	6	20
	3 + 4	7	23.3
	4 + 3	5	16.7
	8	6	20
	9	4	13.3
	10	2	6.7
NCCN risk group	Low	5	16.7
	Intermediate	10	33.3
	High	10	33.3
	Very High	5	16.7
ADT	Yes	15	50
	No	15	50
Variable		Median	Range
Age (year)		70.5	59–86
iPSA (ng/mL)		8.3	4.1–116.1
Positive core (%)		25	5.3–100
ADT period (month)		30	4–112
Prostate volume (cm^3^)		30.5	15.6–94.6
CTV volume (cm^3^)		32.2	17.8–117.5
Rectal wall volume (cm^3^)		21.4	16.1–31.0
Bladder wall volume (cm^3^)		38.8	24.8–62.6

Abbreviations: NCCN, National Comprehensive Cancer Network; ADT, androgen deprivation therapy; PSA, prostate-specific antigen; CTV, clinical target volume.

**Table 2 cancers-16-00403-t002:** Comparison of the dosimetric parameters of the CTV, rectal wall, and bladder wall for the LS, PS, and VMAT plans.

Structure	Dose Metric	LS	PS	VMAT	*p*
CTV	D98 {Gy (RBE), median (range)}	74.9 (73.9–75.3)	75.0 (73.3–76.0)	74.2 (73.0–77.1)	(a) 1.000 (b) <0.001 (c) <0.001
	D50 {Gy (RBE), median (range)}	76.0 (75.9–76.0)	76.0 (75.3–77.1)	75.9 (75.4–76.5)	0.393
	D02 {Gy (RBE), median (range)}	77.3 (76.6–78.5)	77.1 (76.4–79.3)	77.8 (76.5–78.7)	(a) 1.000 (b) 0.009 (c) 0.014
	Dmax {Gy (RBE), median (range)}	79.3 (77.5–82.0)	77.7 (76.7–80.6)	79.5 (77.8–81.0)	(a) <0.001 (b) 1.000 (c) <0.001
	Heterogeneity index {median (range)}	0.029 (0.018–0.049)	0.028 (0.019–0.048)	0.072 (0.047–0.097)	(a) 0.905 (b) <0.001 (c) <0.001
Rectal wall	V10 {%, median (range)}	37.1 (23.2–46.3)	52.0 (31.7–73.5)	70.1 (63.9–87.2)	(a) <0.001 (b) <0.001 (c) 0.060
	V20 {%, median (range)}	29.5 (18.6–36.3)	41.5 (26.2–60.8)	49.3 (40.3–80.8)	(a) <0.001 (b) <0.001 (c) 0.014
	V30 {%, median (range)}	24.4 (15.5–30.5)	35.5 (22.3–50.7)	35.3 (24.2–66.4)	(a) <0.001 (b) <0.001 (c) 1.000
	V40 {%, median (range)}	20.4 (13.0–26.1)	29.2 (18.6–44.1)	26.2 (16.2–51.9)	(a) <0.001 (b) <0.001 (c) 0.014
	V50 {%, median (range)}	16.9 (10.6–22.0)	24.3 (14.6–37.4)	20.2 (9.5–41.6)	(a) <0.001 (b) <0.001 (c) 0.001
	V60 {%, median (range)}	13.1 (8.1–17.8)	18.4 (10.6–29.0)	15.1 (6.1–35.4)	(a) <0.001 (b) <0.001 (c) <0.001
	V65 {%, median (range)}	10.6 (6.5–15.5)	15.0 (8.0–25.4)	12.7 (4.9–32.2)	(a) <0.001 (b) 0.085 (c) 0.002
	V70 {%, median (range)}	7.9 (4.1–12.5)	10.1 (4.2–21.2)	10.1 (3.6–28.5)	(a) 0.014 (b) <0.001 (c) 0.590
	V75 {%, median (range)}	1.7 (0.2–7.7)	0.8 (0–12.2)	3.4 (0.5–16.0)	(a) 0.006 (b) 0.736 (c) <0.001
	Dmax {Gy (RBE), median (range)}	77.2 (75.5–79.2)	75.9 (74.3–78.0)	77.8 (76.8–79.5)	(a) 0.002 (b) 0.072 (c) <0.001
	Dmean {Gy (RBE), median (range)}	17.6 (11.2–22.2)	24.4 (15.5–34.1)	27.5 (21.9–44.1)	(a) <0.001 (b) <0.001 (c) 0.014
Bladder wall	V10 {%, median (range)}	40.8 (24.5–57.0)	54.4 (35.3–73.4)	60.8 (41.6–84.9)	(a) <0.001 (b) <0.001 (c) 0.060
	V20 {%, median (range)}	34.7 (20.5–47.2)	47.5 (31.9–64.7)	50.5 (35.1–66.5)	(a) <0.001 (b) <0.001 (c) 0.590
	V30 {%, median (range)}	30.6 (17.7–42.5)	43.0 (29.2–58.2)	42.0 (27.0–52.4)	(a) <0.001 (b) <0.001 (c) 0.117
	V40 {%, median (range)}	26.7 (15.3–36.7)	39.3 (26.7–52.3)	32.8 (20.7–41.3)	(a) <0.001 (b) <0.001 (c) 0.001
	V50 {%, median (range)}	23.5 (13.1–32.4)	35.3 (23.9–46.8)	26.0 (16.5–35.3)	(a) <0.001 (b) <0.001 (c) 0.001
	V60 {%, median (range)}	20.2 (10.8–28.2)	30.8 (20.9–40.8)	21.7 (13.5–30.1)	(a) <0.001 (b) <0.001 (c) 0.001
	V65 {%, median (range)}	18.4 (9.5–24.8)	28.3 (18.9–38.0)	19.7 (12.2–27.4)	(a) 0.001 (b) <0.001 (c) 0.001
	V70 {%, median (range)}	15.9 (8.0–20.3)	24.0 (15.7–33.8)	17.4 (10.8–24.4)	(a) 0.001 (b) <0.001 (c) 0.001
	V75 {%, median (range)}	9.3 (0.1–14.0)	11.6 (2.1–26.4)	12.1 (4.7–16.6)	(a) <0.001 (b) <0.001 (c) 1.000
	Dmax {Gy (RBE), median (range)}	78.1 (76.5–79.4)	76.8 (75.9–78.8)	79.0 (77.4–79.6)	(a) 0.014 (b) <0.001 (c) 0.004
	Dmean {Gy (RBE), median (range)}	21.8 (12.8–29.2)	31.0 (20.7–41.0)	30.0 (20.0–37.6)	(a) <0.001 (b) <0.001 (c) 0.590

Abbreviations: CTV, clinical target volume; LS, line scanning; PS, passive scattering; VMAT, volumetric modulated arc therapy; Dx, dose received by x% of the organ volume; Dmax, maximum dose; Vx, the relative volume of the organ that received at least x Gy or Gy (RBE); Dmean, mean dose; RBE, relative biological effectiveness. (a) Comparison between LS and PS; (b) comparison between LS and VMAT; (c) comparison between PS and VMAT.

## Data Availability

Research data are available at [https://data.mendeley.com/datasets/bbjj233tn7/draft?a=89aba735-0607-4077-8336-72c6d5b07ffa] (accessed on 26 January 2023).

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
