# Peer review of "Dosimetric Comparison Study of Proton Therapy Using Line Scanning versus Passive Scattering and Volumetric Modulated Arc Therapy for Localized Prostate Cancer"

_cancers, 2024, doi:10.3390/cancers16020403_

Round 1

Reviewer 1 Report

Comments and Suggestions for Authors

1.      In the Introduction section, the manuscript lacks a comparison of methods used in this work with other therapy methods.

2.      Information and demographics of the treated patient are required.

3.      The manuscript is devoid of tables or any information describing the characteristics of the patient's CT or MRI imaging protocols that were used in the study.

4.      In the discussion section, it is encouraged to discuss the benefits and disadvantages of each studied method.

5.      A short discussion about the current and possible future clinical usage of LS, PS, and VMAT should be added to the discussion section. 

6.      The manuscript should be checked for grammatical errors and problems should be resolved.

Comments on the Quality of English Language

The manuscript should be checked for grammatical errors and problems should be resolved.

Author Response

  1. In the Introduction section, the manuscript lacks a comparison of methods used in this work with other therapy methods.

Reply: We appreciate the reviewer’s advice.

There are various curative treatments for prostate cancer, and many kinds of radiotherapy. We have added an overview of different treatment options for localized prostate cancer in the first paragraph of the Introduction section.

"Radiotherapy is one of the radical treatments for patients with localized prostate cancer. The curability of radiotherapy for localized prostate cancer is comparable to that of surgery [1,2]. Various methods of radiation therapy coexist, including X-ray therapy, brachytherapy, proton therapy, and carbon beam therapy. In common with these radiotherapy methods, to achieve both high curability and low rates of toxicity, it is essential to concentrate the dose to the target volume and reduce the dose to surrounding organs at risk (OARs). High-dose X-ray therapy using intensity-modulated radiation therapy (IMRT) is currently considered one of the standard radiotherapies for prostate cancer [3]."

  1. Information and demographics of the treated patient are required.

Reply: Reply: In accordance with a reviewer's suggestion, we made the following three corrections.

  1. The following texts have been newly added to the Results section. "A total of 30 patients were included in the study; risk groups as defined by the National Comprehensive Cancer Network were 5, 10, 10, and 5 patients in the low-, intermediate-, high-, and very high-risk groups, respectively. The median age was 70.5 years. The median prostate and CTV volumes were 30.5 and 32,2ml. Further details are summarized in Table 1. A comparison of the dose distribution between the three treatment modalities for a representative example case is shown in Fig. 1."
  2. More detailed information on patient background has been added in Table 1.
  3. We have added the following patient details in the figure legend for Figure 1. "76-year-old patient with Favorable intermediate-risk prostate cancer (cT2a, GS = 3 + 4, initial PSA = 9.82 ng/ml). Prostate volume = 50.5ml. "

  1. The manuscript is devoid of tables or any information describing the characteristics of the patient's CT or MRI imaging protocols that were used in the study.

Reply: Thank you very much for your suggestion. 

We have added the following description of the planning CT and MRI to the Materials and Methods section.

"The patients’ trunks were immobilized using NB Bord ENB-16 immobilization shells (Engineering System Co., Ltd, Nagano, Japan). Their lower bodies were fixed by using an air mattress (Vac-Lock, CIVCO Medical Solutions, Kalana, IA) to avoid fluctuations around the femoral head during treatment. Computed tomography (CT) imaging was performed using a SOMATOM Perspective CT Scanner(SIEMENS、Munich、Germany), with each patient in the supine position with a full bladder. CT images of 1.25mm thick-ness were obtained. Magnetic resonance images were obtained in 3 mm slices in all patients. The bladder and rectum were aimed in order to maintain the same conditions as in the planning CT. T2wi and T1Gd images were fused to the CT for treatment planning."

  1. In the discussion , it is encouraged to discuss the benefits and disadvantages of each studied method.
  2. A short discussion about the current and possible future clinical usage of LS, PS, and VMAT should be added to the discussion section. 

Reply: We want to answer these two points collectively. In this study, line scanning showed the best distribution and will lead to a reduction of toxicities in the future.

New proton therapy facilities are adopting PBS as typified by line scanning. Although prostate cancer is a disease that takes a long time to evaluate, it is expected that treatment results of PBS will gradually be published, and the results of this study will be supported clinically. Therefore, proton therapy for prostate cancer is expected to gradually shift from the PS method to the PBS method.

On the contrary, proton therapy has weaknesses in cost and availability. It is unlikely that all prostate cancer patients will be treated with PBS in the future, and VMAT is likely to coexist as an essential treatment option. The following points have been newly added to the Discussion section.

"At many facilities, proton therapy is transitioning from the PS method to the PBS method, represented by LS. Due to the indolent nature of prostate cancer, it will take a long time to evaluate new treatments. In the future, however, it is expected that clinical reports will confirm that PBS is associated with fewer toxicities and PBS will become the standard proton therapy method for prostate cancer. On the other hand, proton therapy has weak-nesses in terms of cost and availability. VMAT, the current standard of treatment, is also expected to remain an important treatment option."

  1. The manuscript should be checked for grammatical errors and problems should be resolved.

Reply: Thank you for your advice regarding English in this paper. Before resubmitting the manuscript, we proofread it again and corrected the vocabulary and grammar. The content is unchanged, but we would appreciate it if you could recheck the paper.

Reviewer 2 Report

Comments and Suggestions for Authors

The manuscript under evaluation is a study involving thirty patients diagnosed with prostate cancer that was conducted to evaluate treatment planning using line scanning (LS), PS, and volumetric modulated arc therapy (VMAT). The evaluation considered dose constraints for the clinical target volume (CTV) with D98 ≥ 73.0 Gy (RBE), rectal wall V65 < 17%, and V40 < 35%, as well as bladder wall V65 < 25% and V40 < 50%. The doses for CTV, rectal wall, and bladder wall were computed and assessed using the Friedman test.

The LS technique adhered to all specified dose limitations. In the case of rectal and bladder walls, 10 (33.3%) and 21 (70.0%) patients treated with PS, and five (16.7%) and one (3.3%) patients with VMAT, respectively, did not meet the defined requirements. The volumes of rectal and bladder walls (V10-70) varied widely and were lower with LS compared to PS and VMAT. LS outperformed VMAT across all dose-volume indices for rectal and bladder walls.

From a reviewer's point of view, The LS method exhibited a reduction in rectal and bladder doses when compared to PS and VMAT, suggesting its potential to mitigate toxicities. 

The manuscript is well-written, Figures are well-designed with no errors. and references are well-positioned. I recommend acceptance of this study in its current format. 

Author Response

Reply: We sincerely appreciate your encouraging comments.

We have made several corrections to the paper, as pointed out by other reviewers. The results, direction of discussion, and conclusions remain unchanged, but we would appreciate it if you would check them out.

Reviewer 3 Report

Comments and Suggestions for Authors

The authors present a well written and well performed study. They investigate the dosimetric performance of two proton therapy techniques (linse scanning, LS; pencil beam: PBS) and volumetric modulated arc therapy (VAMT) for the treatment of localized prostate cancer.

There are only 30 patients that were investigated and the follow up time is short. However, proton therapy is a very interesting but not yet widely adapted treatment option. Therefore more data in this area is very welcome. 

Some points need to be clarified:

title: consider including the wording "proton therapy"

abstract good

Results

please provide more information concerning the groups. e.g. compare prostate size; T stage, Gleason Score. 

Discussion: 4.2: subtitle: do you mean urethral instead of ureteral ? and: conclusion of 4.2. : I don't agree. high dosage on the urethra can cause severe damage : e.g. stricutres. Strictures might be clinically more harmful than bladder damage. Please comment. 

4.5: the authors correctly conclude they did not report on late toxicities. In fact this is a problem in these treatement modalities. I accept that they have an other focus here but I would like to motivate the team to looking into late toxities and late functional and oncologic outcome in the future. 

Consider making a statement on cost and availability of these treatment modalities. 

Author Response

Comments and Suggestions for Authors

The authors present a well written and well performed study. They investigate the dosimetric performance of two proton therapy techniques (linse scanning, LS; pencil beam: PBS) and volumetric modulated arc therapy (VAMT) for the treatment of localized prostate cancer.

There are only 30 patients that were investigated and the follow up time is short. However, proton therapy is a very interesting but not yet widely adapted treatment option. Therefore more data in this area is very welcome. 

Some points need to be clarified:

  1. title: consider including the wording "proton therapy"

Reply: In response to a reviewer's suggestion, we would like to change the title of the article to the following; "Dosimetric comparison study of proton therapy using line scanning versus passive scattering and volumetric modulated arc therapy for localized prostate cancer"

  1.  abstract good

- Results

please provide more information concerning the groups. e.g. compare prostate size; T stage, Gleason Score. 

Reply: Thank you very much for your suggestion. We added the prostate volume in Table 1. Also, we have added details on the T stage and Gleason score.

  1. Discussion: 4.2: subtitle: do you mean urethral instead of ureteral ?

Reply: Thank you for finding the mistake. We have corrected the point.

  1. conclusion of 4.2. : I don't agree. high dosage on the urethra can cause severe damage : e.g. stricutres. Strictures might be clinically more harmful than bladder damage. Please comment. 

Reply: As the reviewer pointed out, high doses of the urethra can lead to serious adverse events such as urethral stricture. This point has been reported in several stereotactic radiotherapy studies for prostate cancer.

In light of this point, we have made the following changes to the relevant Discussion section; "For GI toxicities, dose reduction in the urethra is as important as in the bladder. Clinical trials using stereotactic body radiotherapy have revealed a relationship between urethral dose and early and late GU toxicities typified by urethral stricture [43,44]. However, com-pared to traditional whole prostate irradiation, a prospective clinical trial designed to re-duce the urethral dose and improve GU side effects revealed worse biochemical control [45]. In addition, routine urethral dosage reduction also faces numerous technical difficulties [46]. Reducing the dose in the urethra, located approximately in the middle of the prostate, is more challenging than a partial dose reduction in the bladder. Given these data, further research is needed on the necessity and feasibility of urethral dose reduction."

  1. 4.5: the authors correctly conclude they did not report on late toxicities. In fact this is a problem in these treatement modalities. I accept that they have an other focus here but I would like to motivate the team to looking into late toxities and late functional and oncologic outcome in the future. 

Reply: Thank you for your remarks. We know that it takes a long time to evaluate late toxicities after radiotherapy for prostate cancer, especially GU toxicities. We plan to conduct a prospective study to compare treatment efficacies and toxicities between different modalities with follow-ups longer than five years.

The following points have been added to the limitation section; "However, our study has several limitations. The most significant limitation of this study is that this study is an in silico study and did not compare the actual late toxicities that occurred. Long-term follow-up is necessary to evaluate the late toxicities caused by radiation therapy for prostate cancer, especially GU toxicities. Therefore, a prospective trial with a planned follow-up of at least five years is desirable to compare treatment modalities. Considering this study's results, we are currently preparing a prospective trial to compare the different clinical effectiveness of LS, PS and VMAT in prostate cancer patients. "

  1. Consider making a statement on cost and availability of these treatment modalities

Reply: We would like to thank you for addressing a critical point. In this study, the dose distribution of line scanning was favorable and expected to reduce side effects. However, efficacy and side effects are not the only considerations in deciding a treatment option. The cost and the availability of the therapy also influence the treatment choice. It is true that proton therapy, in particular, is generally an expensive treatment at this time and is not available everywhere in the world.

Unfortunately, this study does not examine the cost and availability of each treatment method, so we have added the following text to the limitation section; "Finally, cost and availability, which should be considered for treatment options as well as efficacy and toxicities, were not considered in this study."

Round 2

Reviewer 3 Report

Comments and Suggestions for Authors

The authors have made a substantial effort and have successfully addressed all my remarks and suggestions. Good.